# Anti-Inflammatory Effects of *Mytilus coruscus* Polysaccharide on RAW264.7 Cells and DSS-Induced Colitis in Mice

**DOI:** 10.3390/md19080468

**Published:** 2021-08-20

**Authors:** Xing-Wei Xiang, Rui Wang, Li-Wen Yao, Yu-Fang Zhou, Pei-Long Sun, Bin Zheng, Yu-Feng Chen

**Affiliations:** 1College of Food Science and Technology, Zhejiang University of Technology, Hangzhou 310014, China; xxw11086@zjut.edu.cn (X.-W.X.); 2111926027@zjut.edu.cn (R.W.); sun_pl@zjut.edu.cn (P.-L.S.); 2Key Laboratory of Marine Fishery Resources Exploitment & Utilization of Zhejiang Province, Hangzhou 310014, China; 3Food and Pharmacy College, Zhejiang Ocean University, Zhoushan 316000, China; 2021040@zjou.edu.cn (L.-W.Y.); zhengb@zjou.edu.cn (B.Z.); 4Zhejiang Marine Development Research Institute, Zhoushan 316000, China

**Keywords:** *Mytilus coruscus* polysaccharide, RAW264.7, immunomodulatory, probiotics, ulcerative colitis

## Abstract

Considerable literature has been published on polysaccharides, which play a critical role in regulating the pathogenesis of inflammation and immunity. In this essay, the anti-inflammatory effect of *Mytilus coruscus* polysaccharide (MP) on lipopolysaccharide-stimulated RAW264.7 cells and a dextran sulfate sodium (DSS)-induced ulcerative colitis model in mice was investigated. The results showed that MP effectively promoted the proliferation of RAW264.7 cells, ameliorated the excessive production of inflammatory cytokines (TNF-α, IL-6, and IL-10), and inhibited the activation of the NF-κB signaling pathway. For DSS-induced colitis in mice, MP can improve the clinical symptoms of colitis, inhibit the weight loss of mice, reduce the disease activity index, and have a positive effect on the shortening of the colon caused by DSS, meliorating intestinal barrier integrity and lowering inflammatory cytokines in serum. Moreover, MP makes a notable contribution to the richness and diversity of the intestinal microbial community, and also regulates the structural composition of the intestinal flora. Specifically, mice treated with MP showed a repaired *Firmicutes/Bacteroidetes* ratio and an increased abundance of some probiotics like *Anaerotruncus*, *Lactobacillus*, *Desulfovibrio*, *Alistipe*, *Odoribacter*, and *Enterorhabdus* in colon. These data suggest that the MP could be a promising dietary candidate for enhancing immunity and protecting against ulcerative colitis.

## 1. Introduction

The diversification of the dietary structure is the most direct way to improve quality of life. Digestive system diseases seriously affect the quality of life of patients, and inflammatory bowel disease (IBD) is one of the most common digestive system diseases [1]. Continuous weight loss, recurrent diarrhea, rectal bleeding, unbearable abdominal pain, and increased risk of colon cancer are all clinical features of ulcerative colitis (UC), and its intractability is second to none in IBD disease [2,3]. Because of the poor quality of life and the risk of colon cancer, which have negative effects on the patient´s physical and psychological status, IBD was listed by the World Health Organization as one of the most intractable diseases of the 21st century [4]. Evidence has shown that the occurrence of UC is due to a combination of factors relating to lifestyle, genetic, immune function, and intestinal microbiota, among others. Existing drugs on the market, such as immunotherapy and steroids, can be effective for UC patients, but their inevitable side effects and high recurrence rate are also serious problems that cannot be ignored [5]. Thus, precise dietary prevention is a better solution and needs to be investigated further.

In the physiological response to infection or injury, macrophages have a unique influence in the ongoing process of inflammation [6]. The production of pro-inflammatory mediators and the aggravation of the inflammatory response are inseparable from the work of macrophages, and thus the course of UC is inextricably linked to macrophages [7]. Many pro-inflammatory cytokines, such as tumor necrosis factor-α (TNF-α), interleukin-6 (IL-6), and interleukin-10 (IL-10), are all derived from macrophages, or release lysosome enzymes, which provide multi-directional mediation of the inflammatory response process [8]. As shown in recent studies, the infiltration and activation of macrophages greatly increases the severity of colitis [9]. Nuclear factor-κB (NF-κB) is a paradigm for the signal transduction and gene regulation related to macrophage infiltration and activation [10]. Once activated, NF-κB induces the production of pro-inflammatory cytokines such as TNF-α, IL-10, and IL-6. In view of the potential correlation between inflammation and macrophages, seeking a way to regulate inflammatory cytokine expression and control macrophage activation is very important. Furthermore, in experiments aiming to study anti-inflammatory mechanisms, scholars have widely used lipopolysaccharide (LPS) to stimulate macrophages in inflammation models. The NF-κB signaling pathway is activated after LPS stimulation, causing changes in the expression of related proteins [11,12].

Tens of thousands of microorganisms live in the intestinal tract, thus establishing the intestinal flora. Studies believe that the course of UC is highly correlated with “environmental factors” such as the imbalance of intestinal flora, impaired intestinal mucosal barrier function, and abnormal secondary metabolites [13,14]. Specifically, undesirable variations in the composition of intestinal flora, the ratio changes of the major phyla *Firmicutes/Bacteroidetes*, the dwindling diversity and richness of intestinal microbial community, and the reduction of total short-chain fatty acids (SCFA) in the intestine and other factors can lead to intestinal systemic inflammation [15]. Many scholars and experts have pointed out that prebiotics or probiotics can improve the structure of the intestinal flora of patients with colitis. Promoting intestinal homeostasis through scientific diet management is an indispensable step for alleviating intestinal inflammation. Therefore, a reasonable diet and regulation of the intestinal microbiota represent a high-yield, low-risk disease management idea.

The medicinal value of *Mytilus coruscus* can be traced back to ancient China. According to Li Shizhen’s “Compendium of Materia Medica”, regular consumption of *Mytilus coruscus* can nourish the kidneys [16]. Both immunomodulatory and anti-inflammatory effects can be demonstrated by active compounds extracted from *Mytilus coruscus* [17,18]. *Mytilus coruscus* has strong activity and is widespread in coastal areas. It is also suitable for large-scale artificial breeding. This may be related to the fact that it is a bivalve mollusk [19]. It has been reported in the literature that *Mytilus coruscus* polysaccharide (MP) is a kind of natural polymer that is one of important active ingredients in mussels [20]. In our previous study, the MP was obtained and identified as a kind of alpha dextran structure of polysaccharide (molecular mass: 4.25 × 10^3^ Da). The main chain connection mode is the glycosidic bond of→4)-α-d-Glcp-(1→and the end groups α-d-Glcp-(1→and α-d-Glcp-(1→6)-α-d-Glcp-(1→pass→4,6)-α-d-Glcp-(1→O-6. In a previous study, we found that to a certain extent MP could relieve ulcerative colitis in a mouse model involving dextran sodium sulfate (DSS); nevertheless, its related anti-inflammatory mechanisms have not yet been determined [21]. Based on this, we tried to explore the meaning in depth by carrying out relevant experiments.

In our work, the anti-inflammatory effect and potential action mechanism of MP using both LPS-stimulated RAW264.7 cells and a DSS-induced UC mice model were comprehensively illustrated. For the in vitro experiment, pro-inflammatory cytokine levels and related mRNA expression as well as the related protein expression levels in the NF-κB signaling pathway were determined in LPS-induced RAW264.7 cells. Meanwhile, the effects of MP on intestinal barrier integrity, inflammatory cytokines in serum, and intestinal flora in the DSS-induced UC mice model were also measured. The results of this study may provide a new strategy for developing functional foods using the effective application of MP for the improvement of intestinal inflammation-related diseases.

## 2. Results

### 2.1. MP Promoted RAW264.7 Cell Proliferation

Figure 1 conveys information about the effects of different concentrations of MP on the proliferation of RAW264.7 cells. After treating RAW264.7 cells for 24 h with MP, as compared to the control group, the relative proliferation rates of the treated RAW264.7 cells were all increased, indicating that MP had no toxic effect on RAW264.7 cell growth and had a protective effect on LPS-induced cell damage. Based on this situation, further experiments could be carried out. When the MP concentrations were 100, 300, and 600 μg/mL, the relative proliferation rates of cells were significantly affected by MP (*p* < 0.01), especially at 300 and 600 μg/mL MP. However, MP had no significant effect on RAW264.7 cell proliferation until the MP concentration was over 600 μg/mL (*p* > 0.05), Besides, with the continuous increase in MP concentration, the relative proliferation rate of cells continued to decline. Therefore, we chose 100, 300, and 600 μg/mL as the best concentrations for follow-up research.

### 2.2. MP Reduces the Secretion Level of Inflammatory Factors and the Relative Gene Expression of RAW264.7 Cells

The experiment determined the secretion levels of tumor TNF-α, IL-6, and IL-10 in RAW264.7 cells after treatment with different MP concentrations. RAW264.7 cells were treated for 24 h; subsequently, the supernatant was taken for determination of the secretion of TNF-α, IL-6, and IL-10 with ELISA kits. As shown in Figure 2A, compared to the control group, the secretion levels of TNF-α in the LPS group showed a significant upward trend (*p* < 0.001). However, after treatment with 600 μg/mL MP, the TNF-α secretion level of RAW264.7 cells was significantly decreased (*p* < 0.01). As shown in Figure 2B,C, after 24 h of treatment with LPS, the IL-6 and IL-10 levels in the LPS group showed an identically significant rise (*p* < 0.001) compared with the control group. When adding 100, 300, and 600 μg/mL MP, the levels of IL-6 and IL-10 in RAW264.7 cells were significantly decreased, particularly at 300 and 600 μg/mL MP (*p* < 0.001). According to the above experimental results, MP pretreatment can effectively reduce the LPS-induced increase in the secretion levels of TNF-α, IL-6, and IL-10, with a good dose-dependent relationship, exhibiting a good inflammation-relieving effect in LPS-induced inflammatory RAW264.7 cells.

Figure 2D shows that when compared with the blank control, in RAW264.7 cells given LPS for 24 h the mRNA expression level of TNF-α increased significantly (*p* < 0.001). TNF-α mRNA levels in the MP treatment group were lower than in the LPS-positive control group, showing a remarkably good dose–response relationship. At the same time, at MP concentrations of 300 μg/mL and 600 μg/mL there were no significant differences in the expression level of TNF-α mRNA between the MP treatment group and the positive control group. As shown in Figure 2E,F, the levels of IL-10 and IL-6 in the LPS group were significantly increased compared with the control group (*p* < 0.001). Compared to the LPS group, the levels of IL-10 and IL-6 in RAW264.7 cells were significantly decreased in the groups pretreated with 100, 300, and 600 μg/mL MP (*p* < 0.001). The secretion levels of IL-10 and IL-6 in 600 μg/mL MP group were not obviously higher than those of the blank control group (*p* > 0.05), and the MP group showed good dose-dependence for each index. Above all, MP both showed a good anti-inflammatory effect on secretion and mRNA expression levels in RAW264.7 cells.

### 2.3. MP Regulated the NF-κB Signaling Pathway in RAW264.7 Cells

To study the effect of MP on the NF-κB signaling pathway in RAW264.7 cells, the phosphorylation levels of IκBα and p65 (p-IκBα and p-p65) were examined. As shown in Figure 3, as compared to the blank group, the expression levels of p-p65 (Figure 3B) and p-IκBα (Figure 3C), as well as the ratios of p-p65/p-65 (Figure 3F) and p-IκBα/IκBα (Figure 3G) in RAW264.7 cells treated with LPS alone were significantly increased (*p* < 0.001), showing a severe inflammatory response. After treatment with MP for 24 h, for the expression levels of p-p65 and ratios of p-p65/p-65 with 300 and 600 μg/mL MP showed a significant reduction effect in RAW264.7 cells, and the expression levels of p-IκBα and ratios of p-IκBα/IκBα also declined significantly under conditions of 100, 300, and 600 μg/mL MP. Besides, the decreasing trends of phosphorylated expression levels and relative ratios were enhanced with the increased MP concentrations. Taken together, these results suggested that MP could protect LPS-induced cellular immune regulation by regulating the NF-κB signaling pathway.

### 2.4. MP Ameliorated Inflammatory Symptoms of DSS-Induced Colitis in Mice

In the experiment, a DSS-induced ICR mouse inflammation model was set up to study the effect of MP on colitis. Phenotypic changes in mice were observed and recorded during the experiment. After dissection, it was found that the colon of the mouse was significantly wrinkled and the contents of the colon were black, indicating that the colitis model was successfully constructed (Figure 4C). Body weight changes after 7 days of modeling showed that the weight loss of mice in MP group was much more subtle than that in the DSS group, and the weight changes in LMP and HMP groups showed significant changes compared to the DSS group (*p* < 0.05) (Figure 4A). The disease activity index (DAI) of the MP group was also lower than that of DSS group (Figure 4B). After measurement, it was found that the colon length of mice was shortened to some extent after drinking DSS (*p* < 0.001), but the situation of the MP+DSS group was significantly better than that of the group that received DSS alone, and the colon length and naked eye observations of the MP+DSS group were significantly better than those of DSS group, particularly with respect to the DSS + 600 mg/kg MP group.

### 2.5. MP Improved Oxidative and Inflammatory Damage of DSS-Induced Colitis in Mice

We collected the intestines and serum of colitis mice to detect the degree of oxidative stress. As can be seen from the results of Appendix A, the glutathione peroxidase (GSH-Px) and superoxide dismutase (SOD) levels of the DSS-treated mice were significantly reduced (*p* < 0.01), while compared with the DSS group, the GSH-Px and SOD levels were significantly increased after MP treatment. Inflammatory factors in the intestine and serum were also included in the test, as shown in Appendix A. Compared with the blank control group, DSS could accelerate the secretion of pro-inflammatory factors (TNF-α and IL-6), while the MP group could reverse this phenomenon. It is worth noting that there was no significant difference in the levels of TNF-α and IL-6 in the MP group with different doses in the intestine (*p* > 0.05), while the HMP group showed a significant downward trend in the serum (*p* < 0.05). DSS also could destroy the intestinal mucosal barrier, increasing intestinal mucosal permeability and damaging the intestinal mucosal immune barrier. Compared to the blank control group, the diamine oxidase (DAO) and endotoxin (LPS) levels in the DSS group were significantly increased (*p* < 0.01). After treatment with LMP and HMP, the levels of DAO and LPS were decreased in MP groups, especially HMP (*p* < 0.05) (Figure 5A,B). In general, MP was beneficial for improving the intestinal stress, inflammatory response, and intestinal mucosal immune barrier in DSS-induced mice. Meanwhile, the secretion levels of DAO and LPS in the serum are consistent with the results in the intestinal tract (Figure 5C,D). Combining the above data, we selected HMP as the object of an intervention study of colonic ultrastructure and intestinal flora in the following experiment.

### 2.6. MP Mitigated the Histopathology of DSS-Induced Colitis in Mice

As shown in the transmission electron microscope images (Figure 6A–C) the microvilli in the control (normal) group had the same length and orderly arrangement. The epithelial cell membrane structure was complete, and the tight connection between apical cells was clearly visible. However, the microvilli in DSS group were obviously damaged, disordered, and loose, with different directions, occasional fractures, and an indistinct tight connection. After processing, compared to DSS group, the microvilli in MP group obviously recovered, the arrangement was more orderly, and a tight connection was also faintly visible.

The results of scanning electron microscope in Figure 6D–F further verified the protective effect of MP on the intestinal tract. The microvilli in DSS group were arranged loosely with large gaps, indicating that the intestinal mucosal barrier was damaged. However, the microvilli in the blank group and MP group were basically the same in length, with a small gap. MP treatment improved the microstructure and morphology of the colon in DSS-induced colitis mice.

### 2.7. MP Regulated the Composition of the Gut Microbiota in DSS-Induced Colitis Mice

#### 2.7.1. α Diversity Analyses

The detection depth of intestinal flora can be reflected by the coverage rate. The higher the value, the fewer the undetected sequences in the sample. The results showed that the coverage rate of all three groups was greater than 99.9% (Table 1). α diversity analysis is used to evaluate the abundance and diversity of intestinal flora. Chao1 and Ace are often used as tools to assess abundance. The larger the number, the greater the abundance of intestinal flora. The Shannon index and the Simpson index are used to evaluate diversity. The larger the value, the greater the diversity of flora. Compared to the control group, the values of Chao 1 and Ace in DSS group significantly decreased (*p* < 0.01), while the values of Chao1 and Ace in mice increased after MP gavage, with a significant difference compared to DSS (Figure 7A,B). Compared to the control group, the Shannon and Simpson indexes of DSS groups significantly increased (*p* < 0.01), while the Shannon and Simpson indexes of mice after MP gavage showed a descending trend. (Figure 7C,D).

#### 2.7.2. β Diversity Analyses

A Venn diagram is used to compare the types and quantity of OTUs among different components, indirectly reflecting the similarities and specificities among different samples. As shown in Figure 8A, the number of OTUs in the control group, the DSS group, and the MP group were 629, 695, and 620, respectively. The three groups have 460 over-lapping OTU numbers, of which the number of unique OTUs in the DSS group were 151. However, the number of unique OTUs in the control group and MP group were 38 and 21, respectively. The number of specific OTUs in DSS group was significantly larger than that in control group, while the number of OTUs in MP group was closer to that in control group. Based on the above results, we can draw the conclusion that DSS promotes the occurrence of intestinal inflammation and changes the structure of the intestinal flora in view of the huge survival pressure. It was also clarified that the intestinal flora of mice fed with MP was improved and closer to that of the control group.

Principal coordinates analysis (PCOA) is generally used in microbiome evaluation to analyze intra-group and inter-group differences in samples. As shown in Figure 8B, the obvious separation between the control group and DSS group indicated that the intestinal flora of mice had been greatly changed by DSS, while the MP group samples crossed with the control group, implying that the intestinal flora of mice after MP gavage was similar to that of the control group.

#### 2.7.3. Changes in Intestinal Flora at the Phylum Level

According to the analysis at the phylum level and as shown in the histogram (Figure 9A), the phyla in intestinal tract mainly included *Bacteroidetes*, *Firmicutes*, *Proteobacteria*, *Patescibacteria*, *Actinobacteria*, and *Epsilonbacteraeota*, among which *Bacteroidetes* and *Firmicutes* accounted for the majority. The average proportions of *Bacteroidetes* in the control, DSS, and MP groups were 23.73%, 50.14%, and 38.07%, respectively, and *Firmicutes* in each group accounted for 48.21%, 41.31%, and 38.41%, respectively. According to the gate-heat map (Figure 9B), the abundance of *Firmicutes* in the DSS group was significantly lower than that in control group (*p* < 0.05), and the *Firmicutes* abundance in mice treated with MP was increased. The MP group inhibited the proliferation of *Bacteroides* induced by DSS treatment. Furthermore, *Actinobacteria* showed a significant decline in the DSS group; conversely, the MP group had the ability to reverse this phenomenon.

#### 2.7.4. Changes in Intestinal Flora at the Genus Level

The analysis was carried out at the genus level (Figure 10A,B); we can see that the main bacteria species were *Bacteroides*, *Romboutsia*, *Lactobacillus*, *Desulfovibrio*, *Alistipes* and *Turicibacter*, etc. Among these, the presence of *Bacteroides*, *Romboutsia*, *Anaerostipes*, and *Turicibacter* increased after DSS treatment, and the genera of *Anaerotruncus*, *Lactobacillus*, *Desulfovibrio*, *Alistipe*, *Odoribacter*, and *Enterorhabdus* were less present as compared to the control group. MP intervention could reverse their abundance. It is worth noting that MP intervention could promote the abundance of *Akkmansia* and *Brautia*. Notably, both play a non-negligible role in intestinal health.

## 3. Discussion

The incidence of UC is particularly significant in the presence of unhealthy living habits, certain genetic factors, abnormal immune function, and changes in intestinal flora. Just like the two sides of a coin, clinical drugs have good therapeutic effects, but they are often accompanied by many unavoidable side effects and recurring illnesses. Thus, diet has an unignorable effect on the good function of the body, and it has shown great potential for the improvement of UC. *Mytilus coruscus* polysaccharide (MP), a kind of alpha dextran structure of polysaccharide (molecular mass: 4.25 × 103 Da), is a rich and important active ingredient in mussels. In a previous study, we found that MP can improve DSS-induced UC in mice to some extent; however, its relevant anti-inflammatory mechanism remained unclear. Thus, we attempted to comprehensively explore the inflammatory effect of MP on RAW264.7 cells in vitro and analyze the protective effect of MP on the intestinal barrier of mice with colitis induced by DSS and the regulation of intestinal microbes in vivo through further experiments.

Macrophages play an indispensable role in colitis [22]. Macrophages are present throughout the body’s innate immune response, and act as a “power forward” in the inflammatory response, participate in the first line of body defense, and play an indelible role in inflammation [23]. When pathogenic microorganisms invade the body, they activate themselves by recognizing receptors on the surface of antigens, and clear the pathogen by enhancing phagocytosis, secreting various pro-inflammatory cytokines and anti-inflammatory factors to remove pathogens and control the disease process. Inflammatory cytokines, such as TNF-α, IL-6, and IL-10, are the main factors involved in the regulation of immune response in IBD, and the expression and regulation of these cytokines interact and are mutually coordinated [24].

Tumor necrosis factor (TNF-α) has a classic role in pro-inflammatory cytokines. It has the effect of stimulating neutrophils and macrophages and initiates the secretion of other cytokines. At the same time, it produces appropriate pro-inflammatory cytokines, helping to stimulate the body’s normal immune function and resist infection [25]. However, excessive production of TNF-α will have negative effects on the body, causing different degrees of damage [26]. IL-6 not only plays a pro-inflammatory role, but also has a certain anti-inflammatory effect [27]. IL-10 mainly has an anti-inflammatory effect and controls the reaction degree of inflammation in vivo [28]. As a well-known typical Gram-negative bacterium, the endotoxin LPS has many biological activities, including an immune-enhancing effect. Activation of macrophages by LPS can enhance the production of inflammatory cytokines and mediators such as TNF-α, IL-6, and IL-10 [29]. In this study, MP showed a good proliferative effect on RAW264.7 cells (Figure 1). Most importantly, it significantly reduced the secretion levels of IL-6, TNF-α, and IL-10 after RAW264.7 cells were treated with LPS (Figure 2A–C). Meanwhile, with an increase in MP dose (from 100 to 600 μg/mL), the secretion levels of IL-6, TNF-α, and IL-10 gradually reduced in RAW264.7 cells, indicating MP was helpful for reducing inflammatory response in LPS-irritated RAW264.7 cells. Besides, the cytokine mRNA expression level detected at the molecular level is consistent with the ELISA results (Figure 2D–F), which provides further proof that MP regulates the secretion of cytokines by promoting cytokine gene expression, thus exerting certain immunoregulation functions in LPS-stimulated RAW264.7 cells. To probe into the immunoregulatory mechanism of MP in RAW264.7 cells, the transfer of NF-κB signaling pathway was detected by immunoglobulin. The NF-κB signaling pathway is an important regulator of pro-inflammatory enzymes, and the relative expression of cytokine genes is also its responsibility. It also plays a key role in the transcription of intestinal epithelial cell biology, mucosal inflammation, and infection [30]. On the way, it also contributes to mediating the expression of certain genes involved in cell metabolism and immune response. [31,32]. The activation pathway of NF-κB is ultimately achieved by degrading the phosphorescent IκBα protein and relieving the inhibition of NF-κB [33,34]. After using lipopolysaccharide to stimulate macrophages, IκBα kinase participates in the phosphorylation of p65 to activate the NF-κB signaling pathway [35]. Subsequently, many related genes involved in the regulation of immune responses, including IL-6, TNF-α, and IL-10 [36,37], are expressed. Our results showed that MP treatment not only significantly reduced the ratios of p-p65/p65 (Figure 3F), but also decreased the ratios of p-IκBα/IκBα (Figure 3G), inhibiting the activity of NF-κB signaling pathway and reducing inflammation in vivo RAW264.7 cells. Specific performance outcomes included inhibiting the secretion of TNF-α, IL-6, and IL-10 and downstream mRNA expression of TNF-α, IL-6, and IL-10. The above results were consistent with the previously reported results of Zhang et al. regarding water-soluble polysaccharides from *Arctium lappa* [38].

The results of in vitro experiments allowed us to discover that MP had good anti-inflammatory activity. Based on this, we carried out animal experiments and tried to explore the activity in-depth from an in vivo perspective. UC occurs in the gastrointestinal tract and can occur slowly in the body over a long period of time, with a high likelihood of relapse without scientific management. Intestinal inflammation and damage to intestinal epithelial cells are its main pathological features [39]. In this experiment, the effects of MP on intestinal mucosal permeability, oxidative stress, and changes in inflammatory factors as well as the possible pathogenesis were discussed. When colitis occurs, the colon will produce many oxygen-free radicals, which will damage the intestinal mucosal barrier [40]. SOD catalyzes superoxide dismutase to scavenge free radicals and reduce lipid peroxidation. At the same time, GSH-Px can scavenge H_2_O_2_ and effectively lower the level of · OH to reduce the damage caused by free radicals in the body. In our study, MP not only improved intestinal oxidative stress (GSH-Px and SOD) (Appendix A), but also ameliorated serum GSH-Px and SOD levels (Appendix A), and had a moderate effect on the integrity of intestinal epithelial cells. The content of DAO and LPS in the intestine and blood incarnate the integrity of the intestinal mechanical barrier and the degree of damage [41]. The endotoxin produced by the bacteria represented by Gram-negative bacteria will leak out from the intestine into the blood and participate in the blood circulation of the body, intensifying the inflammatory response [42]. DAO is a hugely active intracellular enzyme mainly found in intestinal epithelial cells. When the intestinal epithelial cells are in an insalubrious state, DAO is released into the intestine and blood [43]. In present study, after treatment with MP the levels of DAO and LPS in intestinal tract and blood were all decreased in MP groups, particularly HMP (*p* < 0.05) (Figure 5A–D), with significant improvements in the intestinal epithelial barrier. Furthermore, we detected the ultrastructure of UC mice colon by TEM and SEM. The microvilli in MP group clearly recovered and the arrangement was more orderly, with a tight connection also faintly visible (Figure 6C). Besides, the microvilli were basically same in length, with a small gap compared to normal mice (Figure 6F). These performances were the result of the combined effect of the improvements in oxidative stress, intracellular enzymes, endotoxins, and inflammatory factors. To a certain extent, TNF-α and IL-6 were generally increased in the pathological part in mice with colitis. Thus, we also detected levels of TNF-α and IL-6 in the intestinal tract and serum of UC mice (Appendix A). The results obtained from the preliminary analysis of in vivo experiments show that MP can improve the secretion of these cytokines in UC mice, consistent with the results of in vitro experiments, as we expected. From the point of view of appearance, with the improvement in inflammatory cytokines, oxidative stress, and intestinal barrier integrity, the disease activity index (DAI), weight, and colon length of UC mice were also improved (Figure 4).

The imbalance in the intestinal flora (caused by changing the intestinal oxidation and metabolic environment in an inflammatory state) is related to the development of UC, which is also one of the reasons for increased intestinal permeability and greatly increased chances of LPS entering the blood. We immediately started testing the composition and diversity of the intestinal flora of all groups of mice. The Chao 1 index and Ace index are usually used to measure community richness, while the Shannon index and Simpson index are used to express community diversity. As shown in Figure 7A–D, the Chao1 and Ace indexes decreased under the influence of DSS, while the Shannon and Simpson indexes increased. After MP treatment, the above results were reversed, and the relative abundance and diversity of intestinal flora were adjusted. We conducted statistics on the OTU results and found that the total OTUs in the DSS group increased significantly and were greater than the control group, and the number of OTUs of all the groups processed by MP was similar to that of the normal group, perhaps because DSS treatment increased the related toxin-producing bacteria and MP treatment attenuated the above bacteria until returning to normal physical conditions (the results of Venn diagram suggest that the microbial composition of the MP group was similar to that of the control group) (Figure 8A). From the perspective of PCOA analysis (Figure 8B), a similar conclusion was verified: there was an obvious separation between the control group and the DSS group, while the MP group samples overlapped with the control group. *Bacteroidetes* and *Firmicutes* are the main strains of intestinal microbiota. In addition, *Actinobacteria* also play a pivotal role in the intestinal tract [43,44]. The essential vitamins, energy, and intestinal maturity environment of the body are inseparable from this huge and sophisticated microbial community that provides the host with many genetic resources [45]. At the phylum level, compared with the control group, the relative abundance of *Bacteroides* in the DSS treatment group increased, while the relative abundance of *Firmicutes* became smaller. This was in accordance with previous research, in which imbalanced characteristics in the intestinal microbiota of colitis were found [46]. The MP intervention repaired the *Firmicutes/Bacteroidetes* ratio, leading to close to normal conditions. A previous study suggested that the *Firmicutes/Bacteroidetes* ratio may have the effect of affecting the production of SCFAs in the intestine, regulating weight and avoiding the unfavorable impact of inflammation on the body [47]. Treatment with DSS might cause a loss of *Actinobacteria* [48]. Notably, the decline in *Actinobacteria* because of DSS treatment was inhibited in the MP group. The above results imply that MP has a regulatory effect on the changes in the composition of the intestinal flora of mice treated with DSS.

We promptly carried out a more in-depth analysis of the gut microbiota at the genus level. *Bacteroides* promotes the secretion of pro-inflammatory cytokines (TNF-α, IL-6), showing a positive correlation. Therefore, once *Bacteroides* decreases, it has a safeguarding effect on inflammatory bowel disease [49,50]. In this dissertation, consistent with previously published papers, the abundance of *Bacteroides* (pathogenic bacteria) in the intestine of a mouse model with DSS was significantly greater [51,52], and our study showed that MP could significantly inhibit this kind of harmful bacterium proliferation. The physiological effects of *Lactobacillus* have been confirmed in many scientific practices and have gradually moved into the public eye. In addition to inhibiting inflammation, *Lactobacillus* also has a positive effect on the intestinal barrier and intestinal mucosal function [53]. *Alistipes* and *Odoribacter* not only play a brilliant role in reducing intestinal inflammation, but also have the function of promoting intestinal maturation [54]. From the above results, we can clearly understand that as compared to the DSS treatment group, the MP group escalated the abundance of *Lactobacillus*, *Alistipes*, and *Odoribacter*. These ameliorative effects were similar to those of a previous study, where a polysaccharide purified from *Arctium lappa* increased the abundance of *Lactobacillus*, *Alitipes*, and *Odoribacter* and improved the intestinal mucosa to restore its function to a nearly normal state [38]. A previous study showed that DSS treatment not only significantly decreased the abundance of *Desulfovibrio* and *Enterorhabdus*, but also increased the abundance of *Romboutsia* and *Tericibacter* [49,55]. In our study, MP treatment had the ability to reverse the above phenomenon. *Anaerotruncus* has been suggested to produce SCFAs in the gut and have anti-inflammatory activity, showing increased levels in UC patients after adopting fecal bacteria transplantation from healthy humans [56]. This is consistent with our experimental results where a reduced *Anaerotruncus* abundance was found in DSS-treated UC mice, and MP facilitated the growth of probiotics such as *Anaerotruncus*. Thus, MP can increase the colonization of probiotics in the intestines, inhibit the excessive reproduction of pathogenic bacteria, and reduce the systemic inflammatory response caused by DSS. The reconstruction in intestinal flora represents a potential anti-inflammatory pathway that may be correlated with the content of SCFAs.

## 4. Materials and Methods

### 4.1. Materials and Reagents

MP was extracted and purified (purity: 85%) by our laboratory. The RAW264.7 cell line was purchased from the cell bank of Shanghai Institute of Cell Biology and Biochemistry, Chinese Academy of Science (Shanghai, China). DSS (MW: 36–50 kDa) was from MP Biomedicals (Santa Ana, CA, USA). Dulbecco’s modified Eagle’s medium (DMEM), fetal bovine serum (FBS), and pancreatin were purchased from Gibco (Grand Island, NE, USA). The Trizol kit, CCK-8 kit, and NF-κB related antibody was purchased from Invitrogen company (Carlsbad, CA, USA). The reverse transcription kit, SYBR Premix Ex Taq (Tli RNaseH Plus), was purchased from Takara (Kyoto, Japan). The ELISA kits required for experiments with IL-6 (EK0411), IL-10 (EK0417), TNF-α (EK0527), and DAO (SBJ-M0211) were purchased from Boshide Biological Engineering Co., Ltd. (Wuhan, China). LPS (055:B5, L2880) was purchased from Sigma (St. Louis, MO, USA). SOD (A001-3-2) and GSH-Px (A005-1-2) kits were purchased from Nanjing Jiancheng Co., Ltd. (Nanjing, China). Primers were designed through Primer Premier 5.0 and synthesized by Sangon Bioengineering Co., Ltd. (Shanghai, China). All other chemicals, solvents, and reagents used were analytically pure.

### 4.2. Cell Culture

RAW264.7 cells were cultured in high-glucose DMEM containing 10% FBS, with 37 °C, 5% CO_2_, and 95% O_2_ as the operating parameters of the incubator. Under normal circumstances, the logarithmic growth phase of RAW264.7 cells is the best period for subsequent experiments. The culture medium was replaced every 1~2 days. The RAW264.7 cells were passaged every 3~4 days at about 80–90% confluence at a 1:3 split ratio.

### 4.3. Cell Proliferation Activity

Collected RAW264.7 cells were seeded into 96-well plates with final density of 4 × 10^3^ cells/well for incubation for 12 h with 5% CO_2_ at 37 °C. After the cells were fully attached, different concentrations of drugs and MP (LPS 100 μg/mL group, and MP 100~1000 μg/mL + LPS group) were added. Besides, the control group (without drugs) and the blank group (without cells) were set at the same time. After 24 h of culture, 10 μL of CCK-8 reagent were added to each well in the dark, with incubation for 1~4 h in dark conditions. When the blank control group reached the appropriate orange-yellow color, the OD value was measured at 450 nm wavelengths using an Auto ELISA detector (Thermo Scientific, Waltham, MA, USA). Cell viability was calculated according to the following formula:(1)$$\mathrm{Relative}\;\mathrm{proliferation}\;\mathrm{rate}\;\left(\%\right)=\frac{\mathrm{OD}\;\mathrm{value}\;\mathrm{of}\;\mathrm{experimental}\;\mathrm{group}-\mathrm{OD}\;\mathrm{value}\;\mathrm{of}\;\mathrm{blank}\;\mathrm{group}}{\mathrm{OD}\;\mathrm{value}\;\mathrm{of}\;\mathrm{control}\;\mathrm{group}}\times 100$$

### 4.4. Determination of Cytokine Secretion Levels of RAW.264.7Cells

RAW264.7 cells were seeded into a 6-well plate with a final density of 3 × 10^5^ cells/well. The cells were cultured overnight and the supernatant was removed according to the following group dosing treatment (1) Blank control group: cultured with complete culture medium containing high-glucose DMEM; (2) Positive control group: 500 ng/mL LPS was added to the complete medium; and (3) MP group: different concentrations of MP (100, 300, and 600 μg/mL) in the presence of LPS were added to the complete medium.

After a certain time of treatment, the cell culture supernatant was centrifuged (1500 rpm, 5 min) to collect clean water. The ELISA kit was used to detect the secretion level of serum interleukin IL-6, IL-10, and TNF-α (please refer to the attached instructions for specific steps).

### 4.5. Determination of Cytokine mRNA Expression of RAW.264.7Cells

RAW264.7 cells were seeded on the 6-well plate with the final density of 3 × 10^5^ cells/well. The cells were cultured overnight and the supernatant was removed according to the cell experiment group described in Section 2.4. After high-glucose DMEM, LPS, and MP treatment, the cells were washed with PBS 3 times. The cells were then collected in an RNASE-free EP tube, and 1 mL of precooled Trizol reagent was added. The cells were repeatedly fitted and lysed, and then stored frozen at –80 °C for further analysis.

According to the IL-6, IL-10, TNF-α, and *β*-actin gene sequences of mice reported by GenBank, the corresponding specific primers were designed based on the Primer Premier 5.0 (Table 2). The cells in the –80 °C refrigerator were taken out, and total RNA was extracted by using TRIZOL one-step method. And the concentration of ODA260/A280, ODA260/A230 and RNA was determined by Nucleic Acid 8 Protein Analysis (Nanodrop, Wilmington, DE, USA), and agarose gel electrophoresis was used to verify the integrity of the extracted RNA (Bio-Rad, Hercules, CA, USA). The general RNA detected was reverse-transcribed into cDNA using a reverse transcription kit.

On adding the corresponding reaction reagent according to the instructions of the Takara fluorescent quantitative kit, the reaction volume was 20 µL, containing 10.0 µL SYBR Green qPCR mix, 0.4 µL ROX II, 2.0 µL cDNA, 6.8 µL dH2O, 0.4 µL upstream primers (10 µM), and 0.4 µL downstream primers (10 µM). The amplification conditions were 50 °C for 120 s, 95 °C for 10 min, and 95 °C pre-denaturation for 15 s, and the corresponding annealing temperature was 60 °C for 40 cycles. Fluorescence collection and preparation of the dissolution curve were in accordance with the instructions of the fluorescence ratio PCR instrument. The expression levels of relative mRNA on every sample target gene were calculated with 2−_ΔΔ_Ct. Three replicate experiments were performed on each sample by quantitative real-time PCR (Applied Biosystems, Forster City, CA, USA), and *β*-actin was used as the reference gene.

### 4.6. Western Blotting

RAW264.7 cells were inoculated into a 6-well plate with a density of 2 × 10^6^/well, and the supernatant was discarded after overnight culture. RAW264.7 cells were treated with different concentrations of MP polysaccharide or LPS (1.0 μg/mL) for 6 h. After washing 3 times with pre-cooled PBS, the cells were collected with cell spatulas in a sterilized EP tube and centrifuged at 1500 r/min for 2 min. The supernatant was discarded and then placed on ice for Western blot analysis. A 10% SDS-PAGE gel was used to separate all the proteins and then transferred to a polyvinylidene fluoride (PVDF) membrane. After being blocked with 5% bovine serum albumin, the membrane was incubated with a 1:1000 dilution of the primary and secondary antibodies. The primary antibodies used in this study were as follows: p-IκBα, IκBα, p-p65, p65, and β-actin. In this experiment, β-actin was selected as an internal parameter to standardize the relative expression of the target protein.

### 4.7. Animals and Experimental Design

The animals (male ICR mice, 6 weeks old) used in this experiment were all provided by Shanghai Slack Laboratory Animal Co., Ltd. (Shanghai, China). The circadian rhythm of the animal room was maintained with the dark/light cycle of 12 h, with a constant temperature of 24 ± 1 °C and humidity of (65 ± 5%). To further illustrate the design of this animal experiment, we developed a brief illustration as shown in the Figure 11 below. Roughly speaking, all the male mice were equally and randomly allocated into 4 different groups (*n* = 6), including the following groups: control healthy mice (control), DSS-induced colitis model mice (DSS), low-dose MP intervention-treated mice (LMP), and high-dose MP intervention-treated mice (HMP). From day 1 to day 21, all the mice in LMP group or HMP group were gavage-administered with 300 mg/kg.bw and 600 mg/kg.bw MP, respectively. Likewise, the other mice (control and DSS groups) were orally gavage-administered with the same amount of pure water daily. From day 14 to day 21, the DSS (3.5% *w*/*v*) was supplied in drinking water to all the mice except for the control group for the induction of UC.

### 4.8. Measurement of Oxidative and Inflammatory Damage Levels In Vivo

For the analysis of SOD, GSH-Px, IL-6, TNF-α, LPS, and DAO in colitis-induced colon tissue and serum, colon tissue samples were suspended in lysis buffer and ground by using a homogenizer. The supernatant was collected by centrifugation (10,000 rpm, 20 min, 4 °C). The SOD, GSH-Px, IL-6, TNF-α, LPS, and DAO levels in the supernatant were measured by using ELISA kits under the manufacturer instructions.

### 4.9. Intestinal Morphology Observation in Colitis Mice

After the mice were dissected, the intestinal tract was taken out, washed with PBS, and dried with filter paper for the ultrastructural studies. A section of the intestinal tract without contents was repaired on ice to a size of about 1 mm × 1 mm × 1 mm with a blade. The tissue mass was put into a centrifuge tube filled with 2.5% glutaraldehyde fixed solution and dehydrated with a series of different concentrations of ethanol solutions. Next, a portion of treated colon samples was coated by 3~6 nm of a gold layer using a sputter coater under vacuum for analysis using a scanning electron microscope (SEM, GeminiSEM 300, ZEISS Ltd., Jena, Germany) at an accelerating voltage of 3.0 kV. Then, the other parts of the treated colon tissue were impregnated with epoxy resin to obtain ultra-thin sections, and the samples were stained with uranyl acetate and aluminum citrate for transmission electron microscopy (TEM, Jeol Ltd., Tokyo, Japan) analysis. The acceleration voltage was 80 kV.

### 4.10. Gut Microbiota Analysis by 16S rRNA Gene Sequencing

The feces of each group of mice were stored at –80°C in cryopreservation tubes for 16S rRNA gene sequencing analysis. In brief, after extraction of genomic DNA from our sample, we used 1% agarose gel electrophoresis to detect the extracted genomic DNA. According to the designated sequencing area (V4-V5), the specific primers were synthesized with barcodes. Randomly selected representative samples were used pre-experiment to ensure that most samples could amplify products with appropriate concentrations at the lowest cycle number. The PCR adopted TransGen AP221-02: TransStart Fastpfu DNA Polymerase; PCR machine: ABI GeneAmp^®^ 9700. All samples were obtained according to formal experimental conditions. Each sample had 3 replicates, and the PCR products of the same sample were mixed and then gelled with 2% agarose gel for electrophoresis detection. The AxyPrepDNA gel recovery kit (AXYGEN Company, Central Avenue, Union City, CA, USA) was used to cut the gel to recover the PCR products and was eluted with Tris_HCl for 2% agarose electrophoresis detection. With reference to the preliminary quantitative results of electrophoresis, the PCR products were detected and quantified with the QuantiFluor™ ST Blue Fluorescence Quantitative System (Promega, Madison, WI, USA), and then mixed in corresponding proportions according to the sequencing volume requirements of each sample. The Miseq library was constructed and the Illumina official linker sequence was added to the outer end of the target area by PCR. A gel recovery kit was used to cut the gel to recover the PCR product. Elution was with Tris-HCl buffer for 2% agarose electrophoresis detection. Sodium hydroxide denaturation was used to produce single-stranded DNA fragments. Finally, Miseq sequencing was started, and the fluorescent signal count results were collected in each round to obtain the sequence of the template DNA fragment.

### 4.11. Statistical Analysis of Data

The data were obtained in the experiment using Prism version 7.0 (GraphPad Software, San Diego, CA, USA) and Origin 11.0 version (OriginLab, Redwood, CA, USA) for statistical analysis. Experimental data were expressed as means ± SEM; significant differences were determined by 1-way ANOVA analysis with post hoc independent samples *t*-tests (*p* values of <0.05, <0.01, and <0.001 are indicated by *, **, and ***, respectively).

## 5. Conclusions

The most evident finding from this study is that MP can exert its impressive anti-inflammatory effects in both in vivo and in vitro models. The MP intervention significantly reduced the expression of inflammatory cytokines and inhibited the activation of the NF-κB signaling pathway in RAW264.7 macrophages modeled using LPS. It is worth mentioning that MP relieves some clinical symptoms of colitis by reducing the level of intestinal oxidative stress and protecting the intestinal barrier from damage. In addition, it is precisely due to treatment with MP that the structure and composition of the intestinal flora in colitis mice are also improved. However, the cause-and-effect relationship between the intestinal inflammation and the production of SCFAs still needs to be explored in further studies. Overall, these data suggested that MP might be used as a safe and effective dietary strategy for the prevention and treatment of colitis.

## Figures and Tables

**Figure 1 marinedrugs-19-00468-f001:**
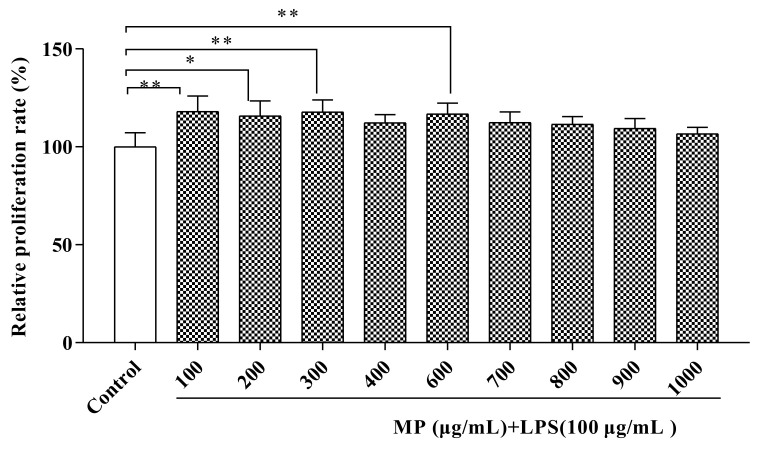
Measurement of potential cytotoxicity of MP in RAW264.7 cells by CCK-8 assay. All the data are presented as means ± SEM. Significant differences of the results were analyzed by 1-way ANOVA analysis with post hoc independent samples *t*-tests (* *p* < 0.05, ** *p* < 0.01).

**Figure 2 marinedrugs-19-00468-f002:**
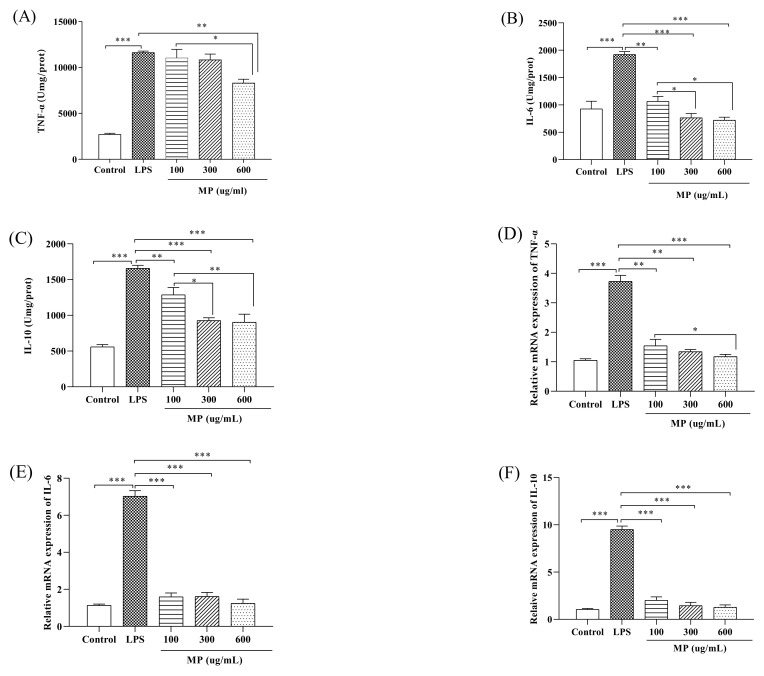
The secretion of TNF-α (**A**), IL-6 (**B**), and IL-10 (**C**) in RAW264.7 cells. The relative gene expression levels of TNF-α (**D**), IL-6 (**E**), and IL-10 (**F**) in RAW264.7 cells. All the data presented are shown as means ± SEM. Significant differences in the results were analyzed by 1-way ANOVA analysis with post hoc independent samples *t*-tests (*p* values <0.05, <0.01, and <0.001 are indicated by *, **, and ***, respectively).

**Figure 3 marinedrugs-19-00468-f003:**
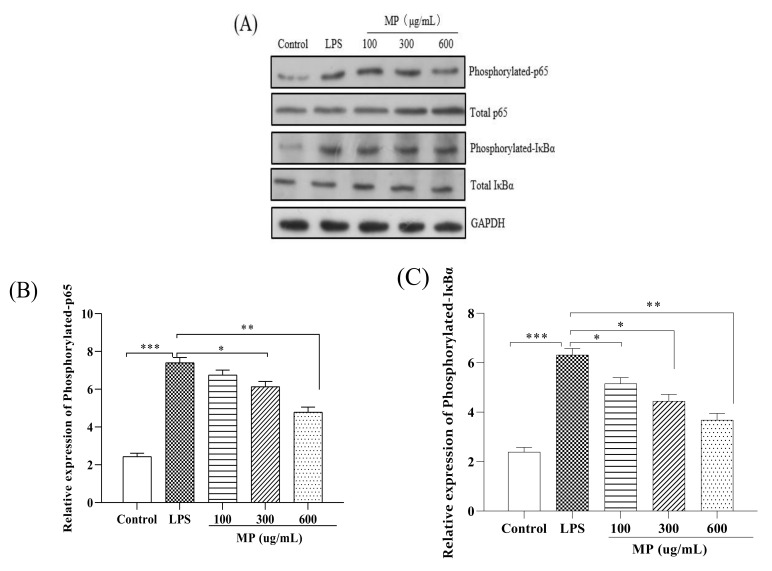
Effect of MP on activation of the NF-κB signaling pathway in RAW264.7 cells. (**A**) Protein levels of p-p65, p65, p-IκBα, IκBα and GAPDH in RAW264.7 cells were investigated using western blot analysis (**B**) Relative expression of p-p65. (**C**) Relative expression of p-IκBα (**D**) Relative expression of total p65 (**E**) Relative expression of total IκBα (**F**) Relative expression of p-p65/p65 (**G**) Relative expression of p-IκBα/IκBα. All the data are presented as means ± SEM. Significant differences in the results were analyzed by 1-way ANOVA analysis with post hoc independent samples *t*-tests (*p* values <0.05, <0.01, and <0.001 are indicated by *, **, and ***, respectively).

**Figure 4 marinedrugs-19-00468-f004:**
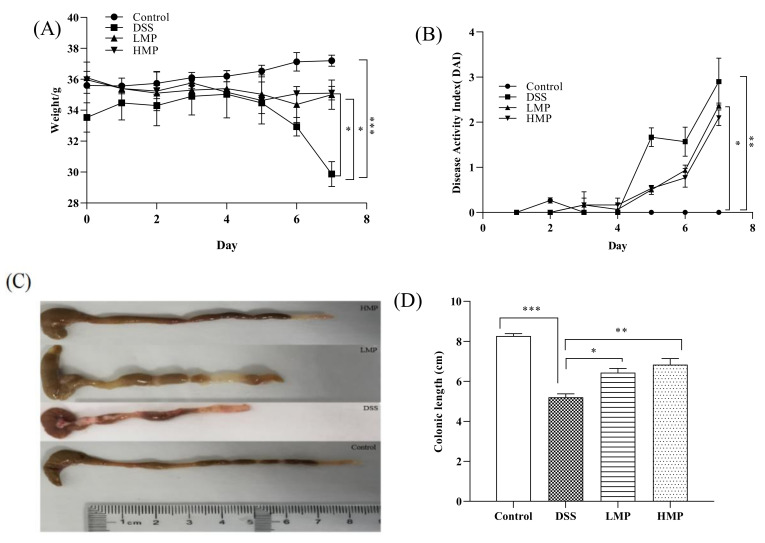
Effect of MP on body weight change (%) (**A**), disease activity index (**B**), macroscopic appearances of colons (**C**), and colon length (**D**) in mice with colitis. Data are presented as means ± SD. Significant differences in the results were analyzed by 1-way ANOVA analysis with post hoc independent samples *t*-tests (*p* values <0.05, <0.01, and <0.001 are indicated by *, **, and ***, respectively). DSS: model group; LMP: DSS + 300 mg/kg MP; HMP: DSS + 600 mg/kg MP.

**Figure 5 marinedrugs-19-00468-f005:**
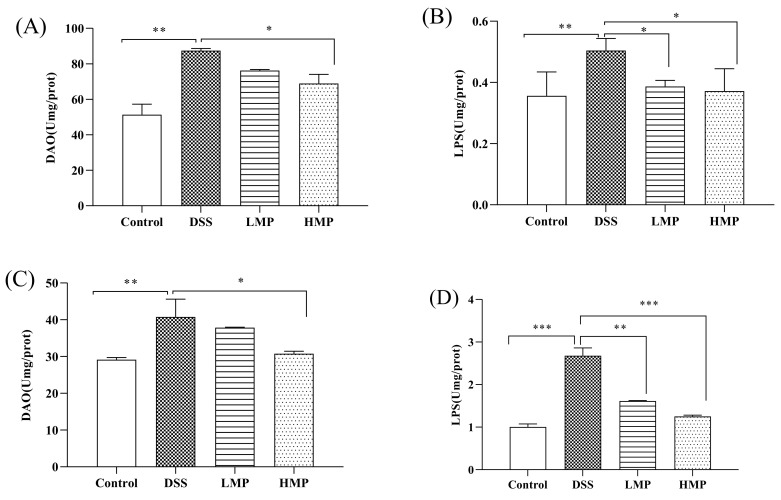
Effect of MP on colonic injury indicators relating to DAO (**A**) and LPS (**B**) in DSS-induced colitis mice, and the effect of MP on serum inflammatory damage relating to DAO (**C**) and LPS (**D**) in DSS-induced colitis mice. All data were presented as means ± SD. Significant differences in the results were analyzed by 1-way ANOVA analysis with post hoc independent samples *t*-tests (*p* values <0.05, <0.01, and <0.001 are indicated by *, **, and ***, respectively).

**Figure 6 marinedrugs-19-00468-f006:**
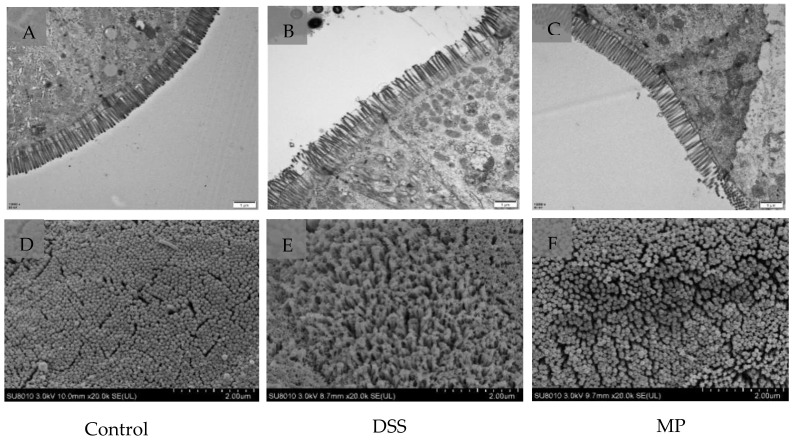
Ultrastructure of mice colon by TEM. (**A**) Control, (**B**) DSS, (**C**) MP (magnification times: 12,000×). Ultrastructure of mice colon by SEM. (**D**) Control, (**E**) DSS, (**F**) MP (magnification times: 20,000×). Control: control group; DSS: model group; MP: DSS + 600 mg/kg MP.

**Figure 7 marinedrugs-19-00468-f007:**
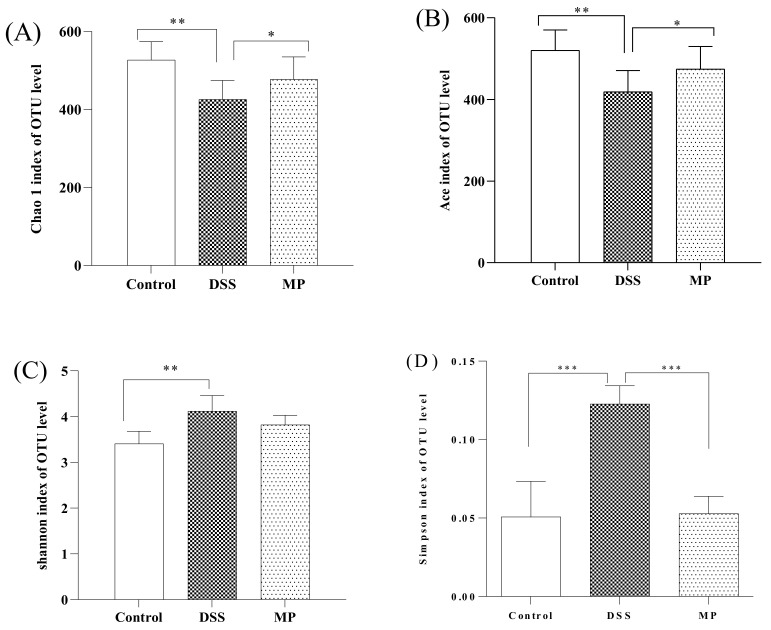
The effects of MP on the alpha diversity of gut microbiota. (**A**) Chao 1, (**B**) Ace, (**C**) Shannon index, (**D**) Simpson index. Control: control group; DSS: model group; MP: DSS + 600 mg/kg MP. All data are presented as means ± SD. Significant differences in the results were analyzed by 1-way ANOVA analysis with post hoc independent samples *t*-tests (*p* values <0.05, <0.01, and <0.001 are indicated by *, **, and ***, respectively).

**Figure 8 marinedrugs-19-00468-f008:**
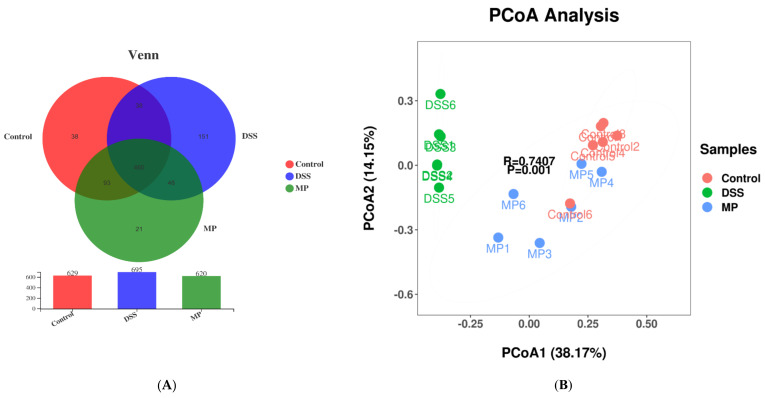
Venn diagram comparison of all the samples (**A**). Plots of PCoA of gut microbiota in the DSS-induced colitis mice (**B**). Control: control group; DSS: model group; MP: DSS + 600 mg/kg MP.

**Figure 9 marinedrugs-19-00468-f009:**
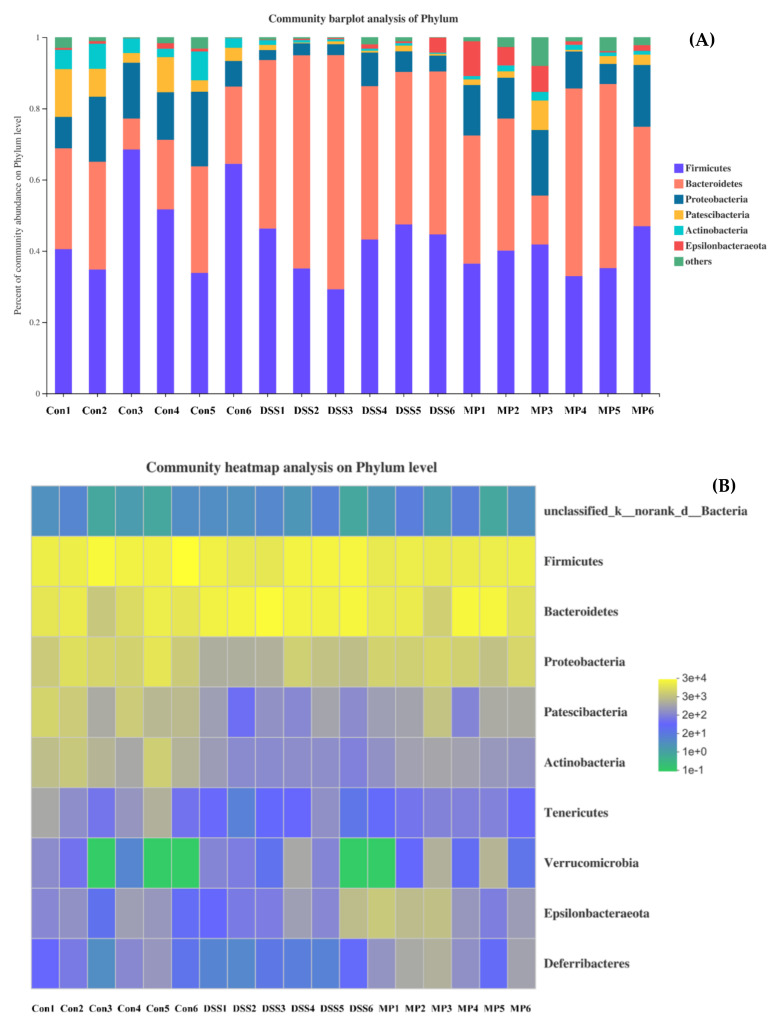
The community structures of the observed samples at the phylum level (**A**). Heat map of all the samples at the phylum level (**B**). Con: control group; DSS: model group; MP: DSS + 600 mg/kg.

**Figure 10 marinedrugs-19-00468-f010:**
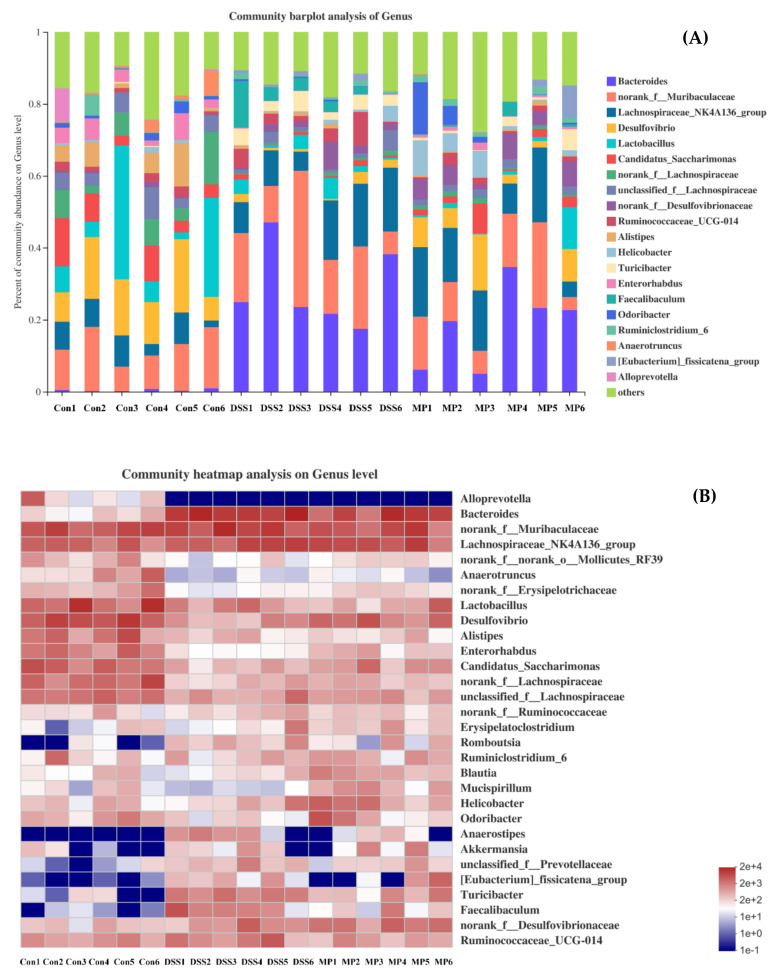
The community structures of the observed samples at the genus level (**A**). Heat map of all the samples at the genus level (**B**). Con: control group; DSS: DSS-treated group; MP: DSS + 600 mg/kg MP.

**Figure 11 marinedrugs-19-00468-f011:**
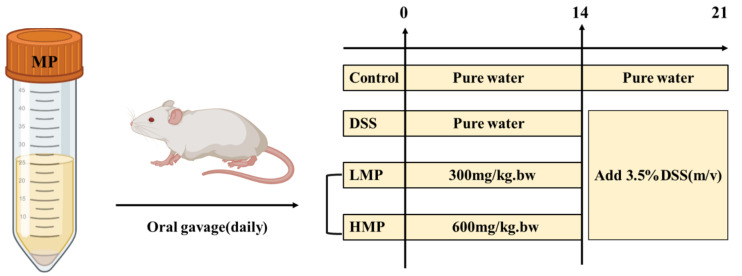
Schematic diagram of animal experiment design for DSS-induced colitis.

**Table 1 marinedrugs-19-00468-t001:** Alpha diversity analysis.

Groups	Good’s Coverage	Chao1	Ace	Shannon Indices	Simpson Indices
Control	>99.9%	527.1 ± 42.87	520.16 ± 45.47	3.48 ± 0.31	0.05 ± 0.02
DSS	>99.9%	425.85 ± 44.37	418.87 ± 47.37	4.11 ± 0.34	0.09 ± 0.03
MP	>99.9%	477.32 ± 51.59	474.36 ± 50.51	3.82 ± 0.20	0.05 ± 0.01

**Table 2 marinedrugs-19-00468-t002:** Oligonucleotide primers used in the quantitative real-time polymerase chain reaction.

Gene	Gene Accession Number	Primer Sequence 5′-3′	Product Size (bp)	Tm (°C)
TNF-α	NM_013693	F:CCCCAAAGGGATGAGAAGTT R:CACTTGGTGGTTTGCTACGA	132	60
IL-6	NM_031168	F:CCAATGCTCTCCTAACAGAT R:TGTCCACAAACTGATATGCT	161	64
IL-10	NM_010548	F:CAGTCGGCCAGAGCCACAT R:CTTGGCAACCCAAGTAACCCTT	144	64
β-actin	NM_007393	F:AGTGTGACGTTGACATCCGT R:GCAGCTCAGTAACAGTCCGC	298	60

## Data Availability

The data presented in this study are available on request from the corresponding author. The data are not publicly available due as public availability violates the consent given by the study participants.

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
