# Peer review of "Anti-Inflammatory Effects of Mytilus coruscus Polysaccharide on RAW264.7 Cells and DSS-Induced Colitis in Mice"

_marinedrugs, 2021, doi:10.3390/md19080468_

Round 1
Reviewer 1 Report
The article entitled "Anti-inflammatory properties of Mytilus coruscus polysaccharide on RAW264.7 cells and DSS-induced colitis mice" is interesting to read and provide worth consider to publish in marine drugs. However, the Authors need to revise major issues in this article before considering to publish in marine drugs.
- Section 4.2- Antibiotics and their concentrations used for this study are missing
- L512 --4×103 => 4×103
- L523, 534 --3×105 =>3×105
- L560 2×106 => 2×106
- L84- ctive => active??
- Some material and methods parts are written in the future tense. Please revise.
- In general, the authors need to carefully revise this manuscript for typos and grammatical issues.
EX- The ELISA kit will be used for testing the secretion level of serum interleukin IL-6, IL-10, and TNF-α, please refer to the attached instructions for specific steps.
- Please provide catalog numbers of ELISA kits
- The efficiency of the primers used in qPCR must be included in the supplementary table.
- Scientific names should be italic - L-81, 85, + + + + +
- In figure 1, please explain why cell viability suddenly decreased by 400 ug/ml.
- Please provide cytoprotective data of MP in presence with LPS.
- If possible, I highly recommend to add NO inhibitory data of MP against LPS stimulation
- L89-92- Please provide the reference for this study
- In figure 2, some control bars have error bars, but some are not showing error bars. Please revise
- According to the results in figure 3, levels of pp65 and pIκBα decreasing in a dose-dependent manner (100, 300, and 600). However, pro-inflammatory cytokine inhibition does not follow the protein expression results given in the western blot analysis. Specifically, proinflammatory cytokine secretion was almost similar at the 300 and 600 treatments. Please explain these results
Reviewer 2 Report
In this manuscript, the authors described in vitro and in vivo anti-inflammatory activities of Mytilus coruscus polysaccharide. They reported the MP could suppress inflammatory activity induced by LPS in RAW264.7 macrophages and ameliorate DSS-induced colitis. Besides, they found that MP could modulate the gut bacterial community and modified gut microbiota may affect the anti-inflammatory activities. It’s an interesting study, however, some questions need to be considered.
Comments
- Line 89-95, the authors described the structure and anti-colitis activities of MP. However, there aren’t any references. If the results haven’t been published, the structure analysis should be added.
- The H&E staining of colon tissue needs to be added to evaluate the ameliorative effect of MP. The tight junction of colon epithelium is more important to check and no shreds of evidence showed effects of MP on intestinal mucosal permeability.
- In the method part, the schematic and description of the DSS model are confusing and inconsistent. Please double-check and simplify them.
- The authors only showed MP group results of the histology and microbiota analysis without indicating which dose of MP. Please revise them.
- The figures didn’t organize well. The font and text size should be consistent. Use Figure 7 as a template.
- The abbreviations should be defined when first time be used in the text. For example, DAO, SOD…
Round 2
Reviewer 1 Report
The authors revised the manuscript as expected and the revised version of the manuscript looks good to consider publishing in marine drugs.
Reviewer 2 Report
Thanks for the professional response. The DSS model is still confusing. Does the DSS treatment last for 14 days by i.p.? It is not a commonly used method for the DSS-induced colitis model. What's the rationale to use this method nor in the drinking water? It's important to explain it in the text.
